# Towards Evaluating Data Priors for Tabular Foundation Models

Zeynep Türkmen [* 1 2]    Kürşat Kaya [* 1 3]    Alexander Pfefferle [4 1]    Frank Hutter [3 4 1]

## Abstract

Data-generating priors are a central component of tabular foundation models because they define the task distribution used during pretraining. However, priors are rarely evaluated as independent components, making it difficult to understand how much they affect downstream model behavior. This raises a methodological question: how can priors from different tabular foundation models be compared independently of the architectures and training protocols they were introduced with? To study this question, we implement a unified interface for publicly available priors from recent tabular foundation models and priors constructed from real datasets. We generate training tasks from each prior, train the same model architecture under a fixed training protocol, and evaluate the resulting models on shared downstream classification tasks. We compare priors through both generated-task statistics and downstream predictive performance. Our results show that different priors favor different downstream behaviors, with some achieving stronger absolute performance and others exhibiting more consistent relative rankings across datasets. We further find that data-level similarity only partially explains downstream behavior. Our code is available at https://github.com/automl/TFM-Playground/tree/prior-dev.

## 1. Introduction

Since the introduction of TabPFNv1 (Hollmann et al., 2023), tabular foundation models have emerged as a promising direction for in-context learning on structured data. These models are pretrained on large collections of tabular tasks, where the data-generating prior defines the task distribution seen during training. As a result, prior design is a central component of tabular foundation models, shaping the downstream predictive behavior of the resulting model.

Despite this central role, priors are rarely evaluated as independent components. Models such as TabPFN, TabForestPFN (den Breejen et al., 2025), and TabICL (Qu et al., 2025) are introduced together with specific choices of prior generation, architecture, optimization, and training protocol. This makes it challenging to isolate the effect of the prior itself. To better understand their role, we take a first step towards exploring how different data-generating priors affect downstream behavior under a controlled experimental setting.

Using this framework, we study whether heterogeneous priors can be compared under a unified setting, whether measurable properties of generated tasks relate to downstream behavior, and how much prior choice matters when architecture and optimization are fixed. Our contributions are as follows: (i) we implement a unified interface for task generation across priors from TabPFNv1, TabICL, TICL (Müller et al., 2025), TabForestPFN, and our own real-data tasks; (ii) we evaluate priors as independent components under a fixed tabular foundation model architecture and training protocol, comparing them through data-level similarity and downstream predictive performance; and (iii) we show that different priors favor different downstream behaviors across datasets, while data-level statistical similarity only partially explains downstream behavior.

## 2. Related Work

Recent tabular foundation models such as TabPFNv1, TabICL, TICL, and TabForestPFN each introduce their own prior-generation mechanisms, described in their respective works. These systems have expanded the range of priors used in practice, but prior design is often treated as a component of the overall model rather than studied in isolation. MITRA (Zhang et al., 2025), on the other hand, provides a more direct study of prior design and task generation. The study proposes a way to identify performance, diversity and distinctiveness of priors and combines priors with these characteristics for model training. MITRA characterizes priors

---

[*]Equal contribution  [1]Department of Computer Science, University of Freiburg, Freiburg, Germany  [2]Zuse School ELIZA, Hochschulstr. 10, 64289 Darmstadt  [3]Prior Labs, Freiburg, Germany  [4]ELLIS Institute Tübingen, Tübingen, Germany. Correspondence to: Zeynep Türkmen <zeyneppturkmen@gmail.com>, Kürşat Kaya <kursat002@gmail.com>.

*Proceedings of the $2^{nd}$ ICML Workshop on Foundation Models for Structured Data*, Seoul, South Korea. 2026. Copyright 2026 by the author(s).

to combine them; we isolate them under a fixed architecture to compare them, and include real-data priors that MITRA does not. Furthermore, other work (Ma et al., 2025) shows that generalization can emerge even from a single real table when diverse tasks are constructed by varying targets and feature subsets.

## 3. Towards a Unified Evaluation Pipeline for Data Priors

To study priors as independent components, we introduce a unified evaluation pipeline that standardizes data generation, model training, and downstream evaluation across methods. We build this work as an extension to the TFM-Playground (Pfefferle et al., 2026). Each prior generates a fixed budget of training tasks used to train an identical nanoTabPFN (Pfefferle et al., 2025) model. Priors are then evaluated through both data-level statistics and downstream predictive performance on datasets drawn from TabArena (Erickson et al., 2025) as shown in Appendix Table 3.

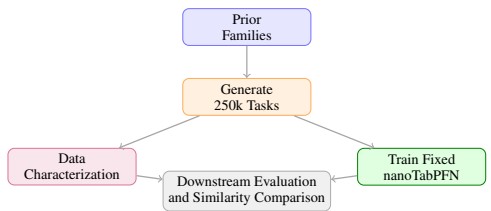

*Figure 1.* Pipeline for studying data priors through generated-data characterization and downstream evaluation.

### 3.1. Unified Prior Generation

We unify priors from multiple libraries under a single interface.

**TabICL:** generates tasks by sampling latent Gaussian variables and propagating them through layered Structural Causal Model (SCM) inspired transformations that introduce dependencies between variables. Across different TabICL variants, layers are parameterized by MLP mappings, tree-based mappings, or a mixture of both. Features and targets are then selected from intermediate variables of the resulting representation space.

**TabForestPFN:** generates tasks by fitting standard machine learning models to randomly sampled data and using their predictions on unseen inputs as targets. Depending on the selected variant, generators are based on decision trees, nearest neighbors, or random projection classifiers, yielding tasks with model-specific decision boundaries.

**TabPFNv1:** uses a prior implementation from a public GitHub repository based on TabPFNv1[1]. It employs func-

tion based priors such as MLPs, Gaussian processes, and mixtures thereof to sample target functions evaluated on random inputs.

**TICL:** uses a closely related function formulation to the TabPFNv1 prior, based on MLP and Gaussian process generators, with alternative preprocessing and target conversion routines.

**Real Data:** uses a controlled real-data task-generation pipeline inspired by recent real-data approaches such as RealTabPFN (Garg et al., 2025) and TabDPT (Ma et al., 2026). The pipeline consists of three stages:

(i) *Dataset curation:* datasets are collected from OpenML, cleaned, preprocessed, and stored together with metadata. (ii) *Leakage reduction:* duplicate and near duplicate datasets are removed by checking against the evaluation pool and within the training pool using statistical fingerprinting based on hashes, dataset dimensions, target statistics, and feature distribution summaries. (iii) *Task construction:* remaining datasets are partitioned into classification, regression, and mixed target pools. Training tasks are generated by sampling datasets, selecting either the original target or a randomly chosen task-compatible column, splitting rows into context and query sets. Task-level preprocessing is then fitted on the context rows only before being applied to both sets.

### 3.2. Experimental Generation Setup

For all priors, we use the default hyperparameters of their publicly available reference implementations. Our goal is not to optimize each prior individually, but rather to compare different priors under a consistent and controlled setup. Future work could examine whether prior-specific hyperparameter tuning changes the relative performance of different priors.

We use a fixed data generation configuration to ensure comparability. Each prior generates 250,000 tasks with up to 512 samples and up to 16 features. For most priors, the feature count is fixed. For the Real Data prior, the number of features and sequence length are capped by dataset availability. For TabICL priors, the minimum feature constraint can behave as a soft constraint, since the original implementation may remove near-constant features during post-generation filtering.

These limits were chosen to keep experiments computationally feasible, particularly as larger sequence lengths and higher-dimensional feature spaces substantially increase memory and runtime requirements. Experiments were conducted with an NVIDIA GeForce RTX 2080 Ti GPU.

---

[1]TabPFNv1 prior repository

### 3.3. Data-Level Prior Characterization

To compare priors at the data level, we summarize generated tabular tasks through a shared vector representation. We use a set of dataset statistics to construct an interpretable summary representation of generated tasks for comparing prior behavior. Each prior is then summarized by aggregating these statistics over many generated tasks. The full definition of the summary vector and its individual statistics is provided in Appendix C.

### 3.4. Controlled Downstream Evaluation

We use nanoTabPFN, a lightweight reimplementation of the TabPFN architecture. Its compact design makes it suitable for controlled and repeatable experiments under limited computational resources while retaining competitive performance in low-data regimes. Because our focus is on the effect of the prior rather than architectural improvements, we keep the model architecture fixed throughout. Architecture and optimization configurations are provided in Appendix A.1.

**Training protocol.** The batch size is set to 1, meaning each gradient step processes a single table. This avoids padding, since the real-data prior produces tables with varying sequence lengths and feature counts. To keep the total sample budget equal across priors, we apply the same batch size across all prior types.

**Evaluation.** Final evaluation is performed after training on a fixed subset of TabArena v0.1 classification tasks. To keep evaluation feasible under our compute budget, we retain tasks with at most 500 features and 5000 instances, resulting in 16 tasks listed in Appendix Table 3. For each task, we use OpenML's predefined train/test splits and evaluate the model across three folds. The resulting prior-vs-dataset ROC-AUC matrix, averaged over folds, is summarized with two heatmaps: a raw heatmap showing dataset difficulty and absolute prior quality, and a per-dataset min-max normalized heatmap highlighting relative prior rankings.

## 4. Results

### 4.1. Dataset Effects and Prior Choice

We first analyze how much downstream behavior changes when only the data-generating prior is varied under a fixed training setup.

Table 1 summarizes downstream performance across TabArena datasets using aggregate metrics. Raw Avg. is the mean ROC-AUC across datasets, and Norm. Avg. is the mean per-dataset min-max normalized ROC-AUC. Wins counts the number of best-performing datasets, while Avg. Rank shows the average rank across all datasets. Because several ROC-AUC differences are small, we focus primarily

| Prior | Raw Avg. | Norm. Avg. | Wins | Avg. Rank |
|---|---|---|---|---|
| `tabicl_mix_scm` | 0.865 | **0.937** | 4 | **2.69** |
| `ticl` | **0.866** | 0.928 | 5 | 2.94 |
| `tabicl_mlp_scm` | 0.864 | 0.920 | 2 | 3.50 |
| `tabpfn_prior_bag` | 0.857 | 0.787 | 0 | 5.50 |
| `tabforest_forest` | 0.855 | 0.770 | 4 | 5.50 |
| `tabpfn_mlp` | 0.852 | 0.746 | 0 | 6.00 |
| `real_random_targets` | 0.844 | 0.675 | 0 | 7.31 |
| `tabforest_neighbor` | 0.846 | 0.670 | 0 | 7.00 |
| `tabicl_tree_scm` | 0.836 | 0.588 | 1 | 6.88 |
| `tabforest_cuts` | 0.833 | 0.475 | 0 | 8.50 |
| `real_default_targets` | 0.781 | 0.123 | 0 | 10.19 |

*Table 1.* Aggregated TabArena classification performance across priors.

on relative profiles and rank-based trends rather than raw averages alone. The full per-dataset normalized performance matrix is provided in Appendix Figure 4.

`ticl` has the highest raw average ROC-AUC and the most dataset wins, indicating stronger aggregate performance across the evaluated datasets. In contrast, `tabicl_mix_scm` achieves the best average rank and strongest relative within-dataset performance under min-max normalization, suggesting more consistent competitiveness across diverse tasks. Overall, these results suggest that prior designs may differ slightly in their performance profiles, but the observed differences should be interpreted cautiously given their small magnitude.

The effect of prior choice also varies substantially across datasets. The per-dataset raw ROC-AUC results (Appendix Figure 3) show that some datasets are more sensitive to prior choice, with larger performance differences across methods. For example, `MIC` exhibits stronger prior sensitivity, whereas `qsar-biodeg` is comparatively stable across priors.

### 4.2. Prior Similarity and Performance Similarity

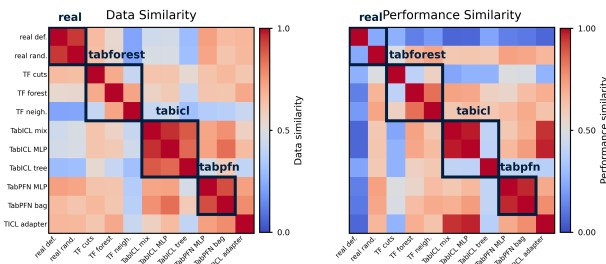

*Figure 2.* Pairwise prior similarity in data space (left) and performance space (right). Data similarity is computed from summary meta-features of generated tasks, while performance similarity is based on normalized TabArena performance profiles. Full matrices are provided in Appendix Figures 5 and 6.

We next analyze whether similarity between generated task distributions translates into similar downstream behavior.

Appendix Figure 5 reveals that priors originating from the same library are generally more similar to each other, suggesting that the generation framework has a stronger effect on the resulting data distribution than the specific underlying mechanism. This pattern is strongest for the TabICL variants, and is also visible for the TabPFN variants, the two real-data priors, and parts of the TabForest family.

Using the real-data priors as a natural reference point, since they are constructed directly from curated OpenML datasets, we observe that among the synthetic priors, `tabpfn_mlp` is one of the closest in data similarity space, suggesting that its generated tasks resemble several measured properties of real tabular data. In contrast, `tabforest_neighbor` is among the most dissimilar under the same summary metrics, yet still achieves competitive TabArena performance (Appendix Figure 3). This suggests that matching measured real-data statistics may be helpful, but is not necessary for strong downstream results.

In performance space, some of the patterns from data space remain visible, but important differences also emerge. Several prior pairs remain close in both spaces, such as the TabPFN variants and the TabICL Mix versus MLP variants, indicating that generated-task statistics may translate into similar downstream behavior. At the same time, notable deviations remain. In particular, `real_default_targets` shows lower similarity to most priors than `real_random_targets`, despite the two being statistically similar in data space. Appendix Figure 4 also shows that `real_random_targets` performs substantially better overall, suggesting that some factors affecting downstream utility are not fully captured by our summary statistics alone.

A likely explanation for the performance difference is task diversity. Because our experiments rely on a limited number of real datasets, repeatedly sampling tasks with the same predefined target can generate many similar prediction problems. In contrast, randomly selecting the target column allows a single dataset to produce multiple prediction problems, increasing diversity and potentially improving generalization, consistent with previous work (Ma et al., 2025). This suggests that random target selection generates a broader variety of tasks.

Overall, data-level similarity is an informative but incomplete proxy for downstream prediction similarity. Priors that look statistically different can still train functionally similar models, while statistically similar priors may differ when they provide different diversity or learning signals.

## 5. Ablation Study for Data Dimensionality

We conducted an ablation study on data dimensionality to examine how feature and sample configurations affect down-stream performance. In our setting, increasing the number of features while keeping the sample size fixed usually led to worse performance on the evaluation datasets. One possible explanation is that these settings create an unrealistic feature-to-sample ratio compared to the downstream evaluation tasks. The selected TabArena classification datasets in Appendix Table 3 have an average feature-to-sample ratio of 0.0350 and a median ratio of 0.0134. Our setup covers four feature-sample configurations: $(10, 200)$, $(30, 200)$, $(50, 200)$, and $(10, 500)$, corresponding to feature-to-sample ratios of 0.05, 0.15, 0.25, and 0.02, respectively. We observe the strongest predictive performance with $(10, 500)$, which is the closest configuration to the evaluation datasets in terms of the feature-to-sample ratio. This ablation complements the prior comparison by showing that generation parameters themselves can shift the induced task distribution and affect downstream performance.

## 6. Conclusion

We present a first step towards a unified evaluation framework for studying data-generating priors as independent components of tabular foundation models. By standardizing prior generation and downstream evaluation while fixing the model architecture and training budget, this setup allows us to investigate how prior choice affects downstream performance.

Across the evaluated priors, we observed meaningful differences in predictive performance and consistency across datasets. No single prior was uniformly best under all criteria: some priors achieved stronger average ROC-AUC, while others showed more stable relative rankings across tasks. This suggests that prior quality depends on the evaluation objective rather than a single aggregate metric.

We further found that data-level similarity only partially explains downstream behavior. Priors with similar summary statistics did not always achieve similar predictive performance. While some statistically different priors had similar average performance. In our real-data setting, increasing task diversity through random target selection was associated with stronger downstream results than relying only on predefined dataset targets. We additionally observed that similarity patterns were often more strongly aligned within prior-generation libraries than across superficially similar mechanisms, such as different MLP-based generators. This suggests that implementation frameworks and task construction choices may substantially shape generated task distributions. This further motivates studying priors more independently from the models they are attached to. We release our code and evaluation setup to support future studies of prior design and to encourage treating priors as first-class components alongside architecture and optimization. Additional limitations are provided in Appendix D.

## Acknowledgement

Funded by the European Union. Views and opinions expressed are however those of the author(s) only and do not necessarily reflect those of the European Union or the European Commission. Neither the European Union nor the European Commission can be held responsible for them. This work was supported by the European Union's Horizon Europe research and innovation programme under grant agreement No 101214398 (ELLIOT).

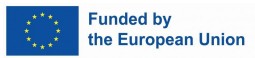 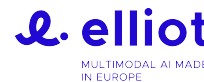

Frank Hutter acknowledges the financial support of the Hector Foundation.

Zeynep Türkmen is supported by the Konrad Zuse School of Excellence in Learning and Intelligent Systems (ELIZA) through the DAAD programme Konrad Zuse Schools of Excellence in Artificial Intelligence, sponsored by the Federal Ministry of Education and Research.

We thank the reviewers for their feedback.

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

# A. Experimental Setup

## A.1. Model and Training Configuration

| Model | | Training | |
|---|---|---|---|
| Layers | 6 | Optimizer | AdamWSF |
| Attention Heads | 6 | Weight decay | 0 |
| Embed. dim | 192 | LR | $10^{-4}$ |
| MLP hidden | 768 | Steps | 250k |
| Loss | Cross entropy | Batch size | 1 |
| Output | Class logits | Grad. clip | 1.0 |

*Table 2.* Shared nanoTabPFN architecture and training configuration used across all priors. Training uses the AdamWScheduleFree optimizer (AdamWSF) (Defazio et al., 2024).

## A.2. Evaluation Datasets

| Task ID | Dataset ID | Dataset Name | # Features | # Samples |
|---|---|---|---|---|
| 363614 | 46906 | anneal | 39 | 898 |
| 363621 | 46913 | blood-transfusion-service-center | 5 | 748 |
| 363623 | 46915 | churn | 20 | 5000 |
| 363626 | 46918 | credit-g | 21 | 1000 |
| 363629 | 46921 | diabetes | 9 | 768 |
| 363671 | 46927 | Fitness_Club | 7 | 1500 |
| 363674 | 46930 | hazelnut-spread-contaminant-detection | 31 | 2400 |
| 363682 | 46938 | Is-this-a-good-customer | 14 | 1723 |
| 363684 | 46940 | Marketing_Campaign | 26 | 2240 |
| 363685 | 46941 | maternal_health_risk | 7 | 1014 |
| 363696 | 46952 | qsar-biodeg | 42 | 1054 |
| 363700 | 46956 | seismic-bumps | 16 | 2584 |
| 363702 | 46958 | splice | 61 | 3190 |
| 363704 | 46960 | students_dropout_and_academic_success | 37 | 4424 |
| 363707 | 46963 | website_phishing | 10 | 1353 |
| 363711 | 46980 | MIC | 112 | 1699 |

*Table 3.* Selected TabArena (Erickson et al., 2025) classification tasks and dataset statistics.

# B. Additional Results

## B.1. Per-Dataset TabArena Results

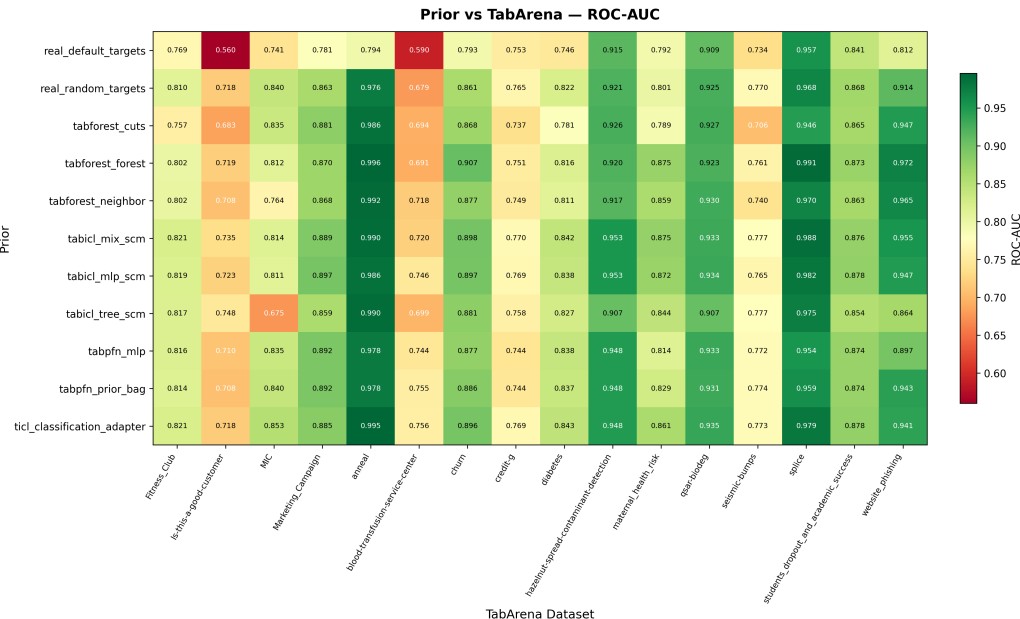

*Figure 3.* Per dataset TabArena ROC AUC across priors. This matrix reports the raw downstream performance values used to analyze dataset difficulty and prior sensitivity.

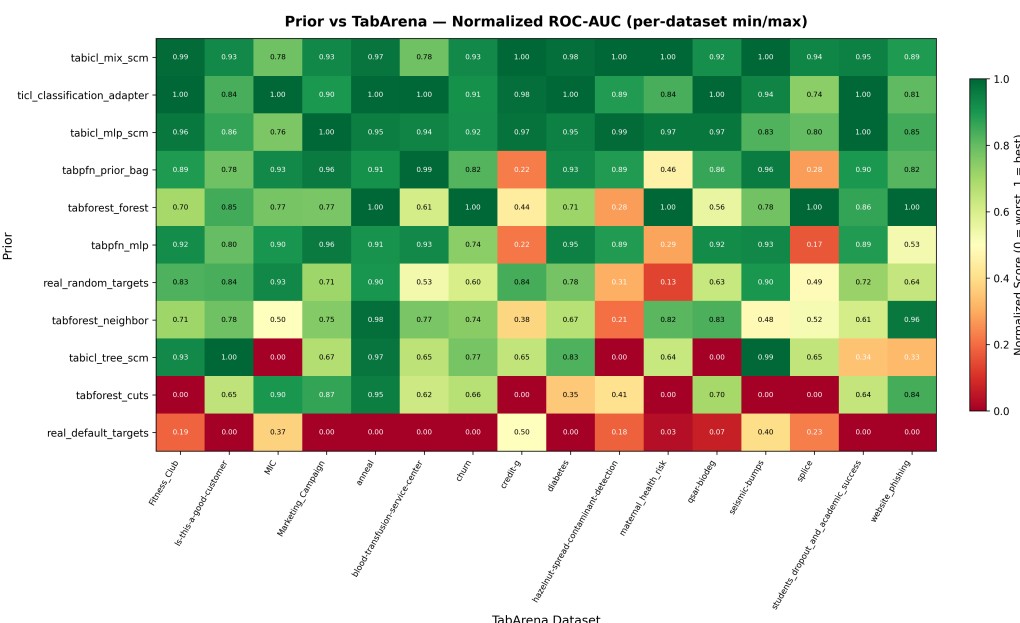

*Figure 4.* Per dataset min-max normalized TabArena ROC AUC across priors. Values are normalized within each dataset to emphasize relative prior performance and dataset specific sensitivity.

## B.2. Prior Data Similarity

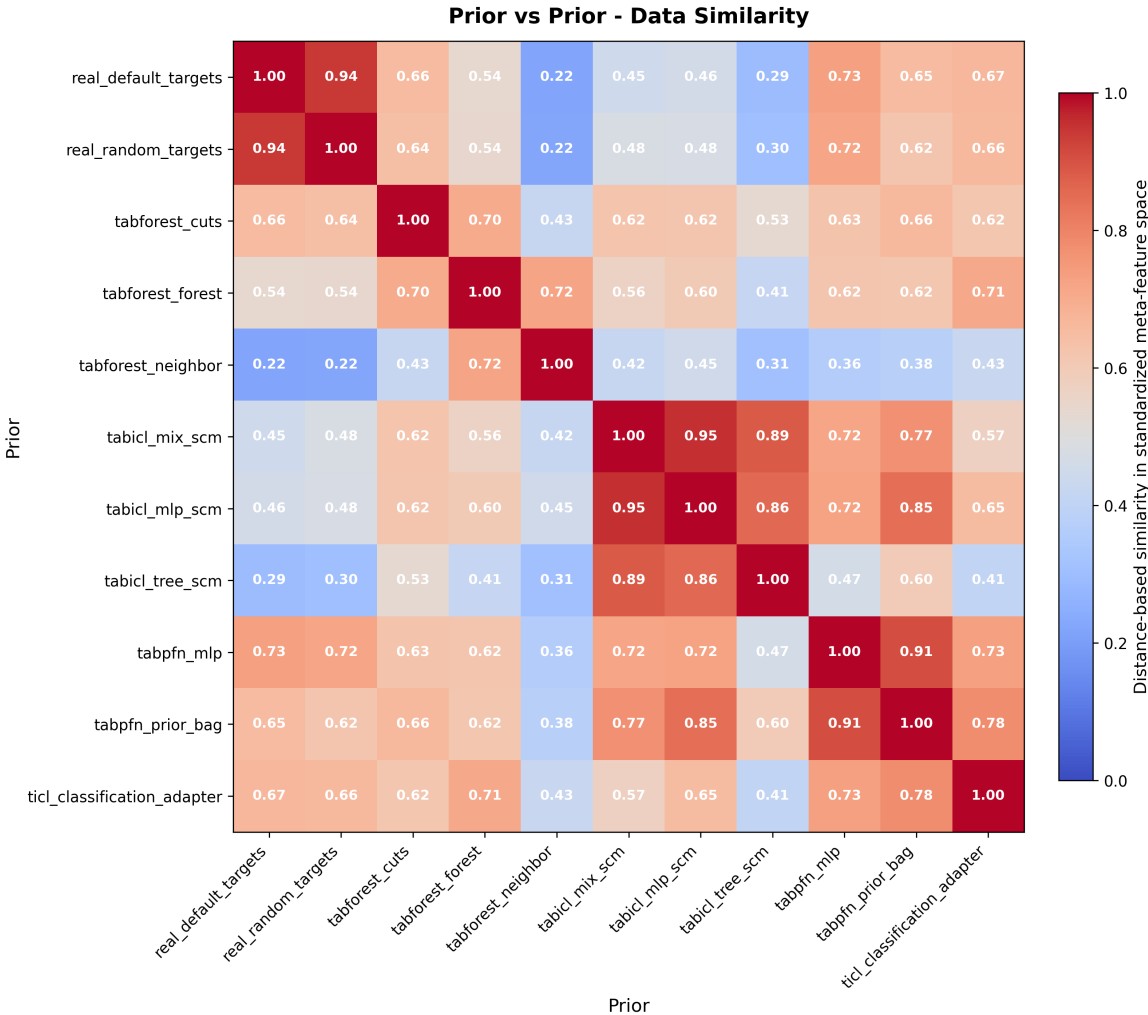

*Figure 5.* Pairwise prior similarity in data space. The reported values are unitless similarity scores derived from distances between aggregated prior summary statistics and mapped to a 0 to 1 scale, where larger values indicate greater similarity in generated task distributions.

## B.3. Prior Performance Similarity

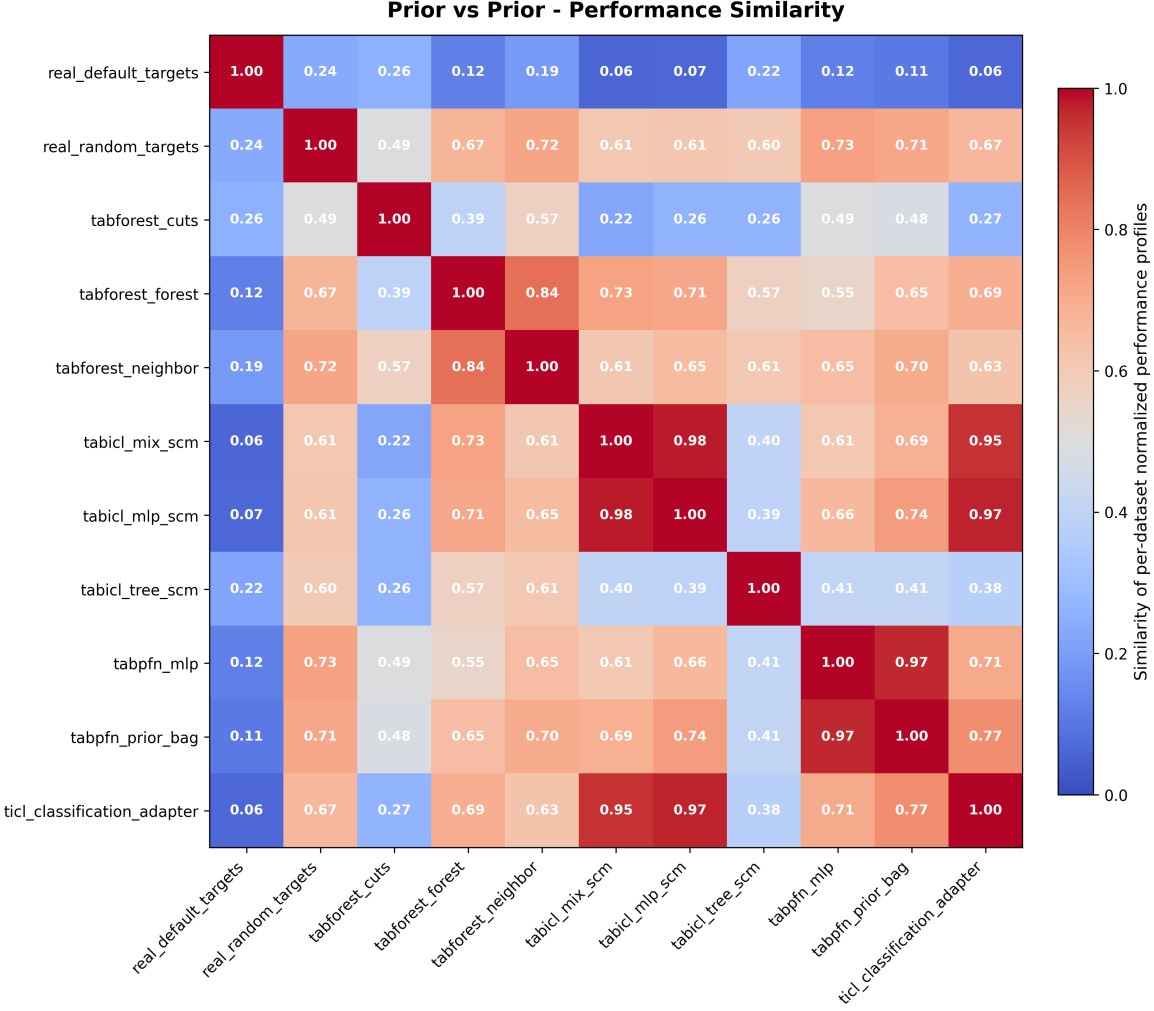

*Figure 6.* Pairwise prior similarity in performance space. The reported values are unitless similarity scores computed from distances between per-dataset normalized TabArena performance profiles and mapped to a 0 to 1 scale, where larger values indicate more similar relative downstream behavior across datasets.

## B.4. An Ablation Study for Data Dimensionality

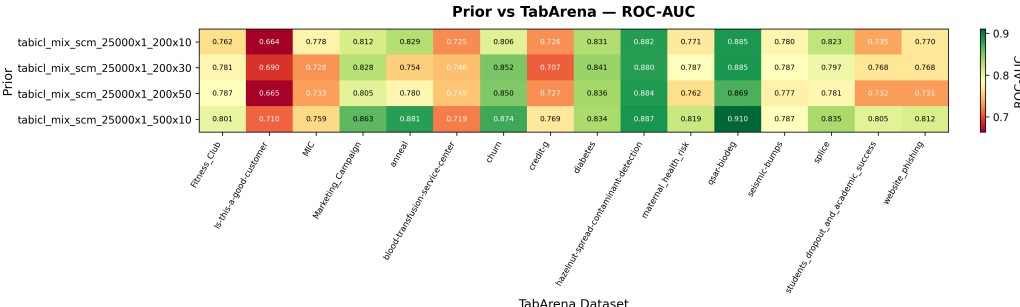

*Figure 7.* TabICL Mixed TabArena dataset evaluation performance with different data dimensionalities

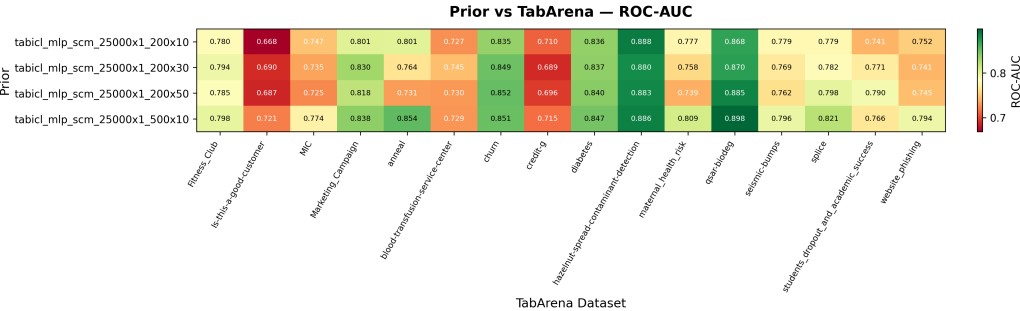

*Figure 8.* TabICL MLP TabArena dataset evaluation performance with different data dimensionalities

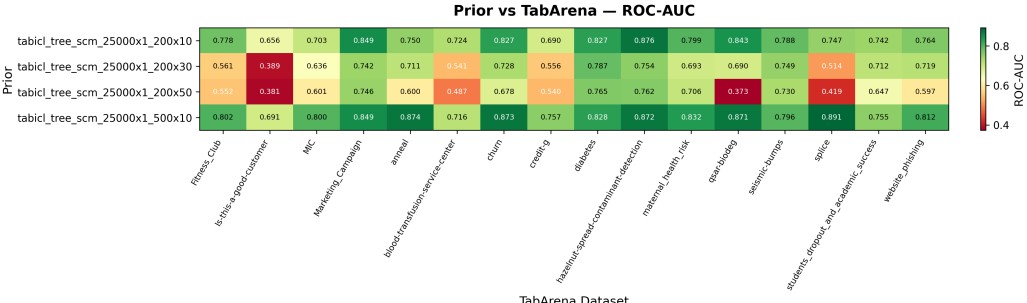

*Figure 9.* TabICL Tree TabArena dataset evaluation performance with different data dimensionalities

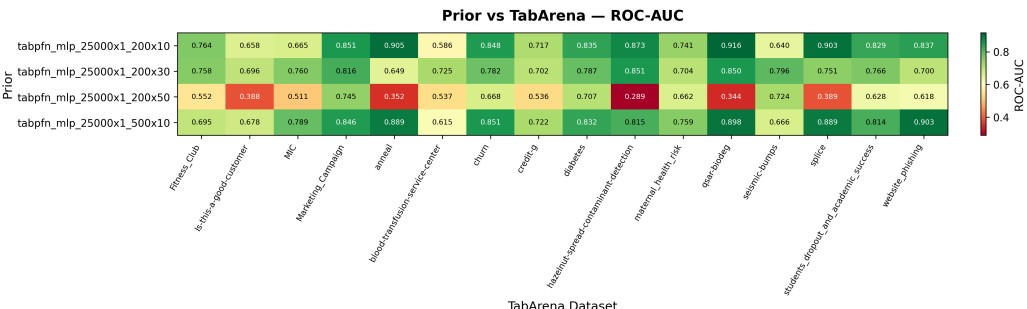

*Figure 10.* TabPFN MLP TabArena dataset evaluation performance with different data dimensionalities

# C. Dataset-Statistic Summary Vector

To compare priors in data space, we summarize generated tasks using a fixed vector of dataset statistics. The purpose of this representation is not to fully describe a tabular task, nor to claim that these statistics are an optimal embedding of tabular data. Instead, we use them as an interpretable proxy for testing whether simple statistical properties of generated tasks can reveal meaningful differences between prior families.

This diagnostic assigns higher similarity to priors that generate tasks with similar label structure, feature-label relationships, feature geometry, and distributional properties. Since data-generating priors define the tasks seen during pretraining, such properties may influence the learning dynamics of the resulting model. We therefore construct a summary vector that captures several complementary aspects of classification tasks.

**Label structure.** We first characterize the target distribution using the number of classes, the majority-class ratio, and the entropy of the class distribution. These metrics capture basic task structure: the number of classes reflects prediction complexity, the majority ratio captures class imbalance, and entropy provides an information-theoretic measure of label balance. These quantities are computed per task and then aggregated, avoiding direct pooling across tasks with different class counts.

**Linear feature-label signal.** We include ANOVA F-statistics and standardized mean differences to capture linear class separability. ANOVA F-statistics measure whether feature means differ across classes relative to within-class variation, while standardized mean differences provide an effect-size view of class-conditional separation. We use both mean and median summaries of the F-statistics: the mean captures the presence of strong discriminative features, while the median is more robust to outliers. Since F-statistics can vary over several orders of magnitude, we apply a $\log(1 + x)$ transformation before aggregation.

**Nonlinear feature-label signal.** To capture dependencies not reflected by linear mean differences, we compute mutual information between features and labels. Mutual information can detect nonlinear or non-monotonic associations that ANOVA-based measures may miss. We summarize both the mean mutual information and the upper quartile, where the latter captures whether a task contains a small subset of highly informative features. In addition, we include a nonlinear-gap statistic, defined as the fraction of features with high mutual information but low ANOVA F-statistics. This provides a heuristic measure of signal that is informative but not well explained by simple linear class separation.

**Feature-space geometry and redundancy.** We characterize the geometry of the feature space using mean absolute feature correlation, variance explained by the first principal component, and the effective rank ratio. Mean absolute correlation captures feature redundancy and collinearity. The top principal-component variance measures whether most variation is concentrated along a single direction. The effective rank ratio provides a normalized estimate of intrinsic dimensionality based on the feature covariance spectrum. Together, these metrics help distinguish priors that generate independent high-dimensional features from those that generate redundant or low-dimensional feature structures.

**Feature distribution shape.** We also include marginal feature-shape statistics. Median excess kurtosis captures whether features are approximately Gaussian-like, heavy-tailed, or spiky. The discrete-feature ratio measures the fraction of features with at most ten unique values, distinguishing continuous feature distributions from more categorical or thresholded ones. These statistics are useful because different priors may generate feature spaces with substantially different marginal distributions, even when their label structure appears similar.

**Operational separability.** Finally, we include the in-sample balanced accuracy of logistic regression. This metric provides an operational summary of how easily the generated task can be solved by a simple linear classifier. We use in-sample performance intentionally: the goal is not to estimate generalization performance, but to characterize the separability structure of the generated data itself. A high value indicates that the generated labels are close to linearly separable, whereas a low value suggests more complex or weaker decision structure.

Overall, the resulting summary vector covers five broad axes of generated classification tasks: label structure, linear signal, nonlinear signal, feature geometry, and feature distribution shape. After computing the summary vector for each prior, we standardize each statistic across priors, compute pairwise Euclidean distances, and transform the distances with an RBF kernel to obtain a bounded similarity score. This gives a simple and interpretable way to compare priors in data space.

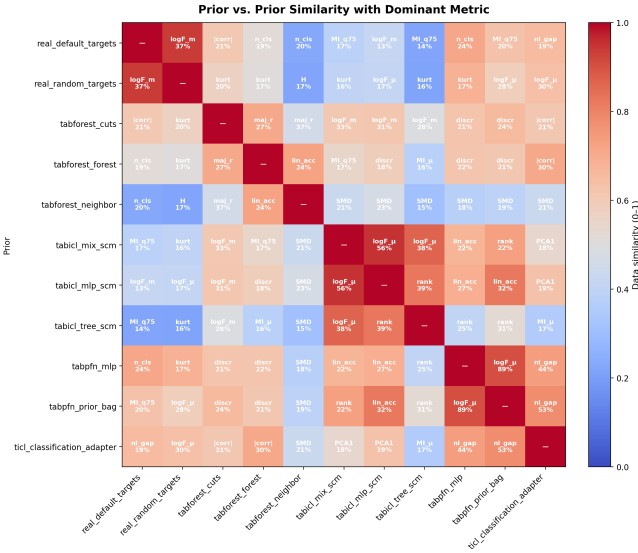

*Figure 11.* Pairwise data-level similarity between priors based on the dataset-statistic summary vector. Cell annotations indicate the statistic contributing the largest share to the standardized squared distance between each pair of priors, with the corresponding percentage shown below.

However, the representation remains incomplete: it does not fully capture higher-order feature interactions, class-conditional feature distributions, or all forms of decision-boundary complexity. For this reason, we interpret data-level similarity as an exploratory diagnostic rather than a definitive measure of prior quality.

**Metric abbreviations.** $n_cls$ denotes the number of target classes. $maj\_r$ is the fraction of samples in the majority class. $H$ denotes the Shannon entropy of the class distribution. $\log F_\mu$ and $\log F_m$ denote the mean and median log-transformed ANOVA F-statistic across features, respectively. $MI_\mu$ and $MI_{q75}$ denote the mean and 75th percentile of feature-wise mutual information with the target. SMD denotes the median standardized mean difference between class centroids across features. $|corr|$ is the mean absolute pairwise feature correlation. PCA1 denotes the fraction of variance explained by the first principal component. Rank denotes the effective rank of the feature matrix normalized by the number of features. $nl\_gap$ denotes the gap between a nonlinear and a linear baseline, used as a proxy for nonlinear structure. Kurt denotes the median feature kurtosis. Discr denotes the fraction of features with few unique values. $lin\_acc$ denotes the training balanced accuracy of a linear classifier.

## D. Additional Limitations

Our study has several limitations. We focus on classification tasks, which enables consistent evaluation but limits direct generalization to regression or other settings, although a regression pipeline is available in the provided repository. All experiments use nanoTabPFN as a single compact backbone, which helps isolate prior effects but does not capture interactions with alternative model architectures or larger scale models. We also evaluate priors under fixed training budgets and a bounded evaluation set, so conclusions may change under broader task collections or different compute regimes.

To preserve comparability, we use the default settings of publicly available prior implementations rather than performing per prior hyperparameter optimization. As a result, some priors may be under or over estimated relative to their best achievable performance. In addition, several priors originate from different codebases, so observed differences may reflect both underlying data generation mechanisms and implementation specific choices such as preprocessing, sampling procedures, or computational efficiency.

Future work could extend this framework to larger models, broader task collections, learned mixtures of priors, richer real data task construction, and stronger measures of prior quality that better predict downstream utility. We release our evaluation pipeline to support reproducible and extensible future studies of prior design for tabular foundation models.

