# OpenReview forum: "Towards Evaluating Data Priors for Tabular Foundation Models"
_ICML.cc/2026/Workshop/FMSD — FMSD @ ICML 2026 Poster_

### Official Review · Reviewer_LNrK · 2026-05-13
**Towards Evaluating Data Priors for Tabular Foundation Models**

**Rating:** 7
**Confidence:** 3

**Review:**

## Summary
The author inspects the influence on the model performance from the distribution of training prior.  Several pretrain prior-generation mechanisms are considered during the study, alongside several real-world datasets. From the experiments, the author observed that different priors influence the downstream performance significantly.
## Strength
1) This is a data-centric study, which is a very important aspect that the community is currently focusing on.

2) The data being composed are comprehensive, including several representative prior-generated data and real-world datasets.

3) The experimental setup is well-written and clearly presented.
## Areas for Improvement
1) The current study looks more like a report of the model behaviors on existing priors. Additional experiments building on top of the current framework including mixing some of the prior together, or mixing the prior using a similarity filter based method, could boost the soundness of the work significantly.

2) The prior naming is a little bit confusing, a table in the appendix describing each shortcut would be helpful.

## Detailed Comments
The detailed suggestion is stated in section *Areas for Improvement*.
## Justification of Scores
The topic of the work nicely aligns with the theme of the workshop titled “Foundation Model for Structured Data”. The problem setup and experimenting scenarios are comprehensive, with space of improvements as stated in the review above.

---

### Official Review · Reviewer_XMtJ · 2026-05-16

**Rating:** 7
**Confidence:** 4

**Review:**

## Summary

This paper proposes an experimental framework to compare various data-generating priors used to create tasks for tabular foundation model pre-training. The authors pre-train the nanoTabPFN foundation model using a selection of data priors from recent literature. Evaluation is then performed on TabArena classification tasks. The experimental results reveal a performance ranking of these priors, as well as measurable similarities between some of them.

## Strengths

* The motivation is clear. To the best of my knowledge, a systematic comparison of data-generating priors in this context has not been performed before, making this a novel and timely contribution to the field.
* The experimental setup is sound and well-designed.

## Areas for Improvement

* Because the performance gap between the different priors is relatively small, additional experiments are needed to establish the consistency of the priors' rankings (see Detailed Comments).
* The paper would benefit from more actionable recommendations. For instance, investigating whether a combination of certain priors could improve overall downstream performance would be an interesting and valuable addition.
* Including more methodological details would increase the paper's accessibility. Currently, readers who are not highly familiar with tabular foundation models might find the paper difficult to follow.

## Detailed Comments

* It would be highly beneficial to include experiments regarding model and data scaling laws. This would provide a more profound analysis of how consistent the performance of each prior is across different model architectures and varying pre-training data sizes.

## Justification of Score

I did not find any major flaws in the methodology or execution. I recommend acceptance.

---

### Official Review · Reviewer_82Yw · 2026-05-20
**Towards Evaluating Data Priors for Tabular Foundation Models Review**

**Rating:** 7
**Confidence:** 4

**Review:**

## Summary
The paper covers an important topic in tabular foundation models (TFMs) on how to evaluate the data priors, which power

## Strengths
- The authors identify a key core question in evaluating TFMs in analyzing the synthetic data priors.
- Sufficient TFM related work is discussed, e.g., TabPFN, TabICL, Mitra and related model families.
- Including real-data priors is a benefit.
- Good use of state-of-the-art tabular dataset TabArena for the evaluation
- Nice general approach by focusing on the data which is model agnostic
- Results are thorough

## Areas for Improvement
- The motivation of studying the prior in isolation is similar to Mitra's. While important and there is more work to do there, it would be better to better differentiate a bit.

## Detailed Comments
- How does it differ from Table 1 in Mitra where each row is the model trained on the individual prior and the columns are tested on data from those individual priors and from the real-world data TabRepo, i.e., in component (ii)?
- How much gain do you get from the real data priors?
- Why is batch size set to 1? Is there a way around that?
- Is the conclusion that the prior selection is still dataset dependent - how does that impact a true TFM then?

## Justification of Score
Overall, I think this paper is very well-suited for the workshop. It aims to address an open question in TFM on the effect of the synthetic priors which is a critical component. They provide a systemic study to explain the effect of each prior one-by-one.